# Palmitoleoylethanolamide Is an Efficient Anti-Obesity Endogenous Compound: Comparison with Oleylethanolamide in Diet-Induced Obesity

**DOI:** 10.3390/nu13082589

**Published:** 2021-07-28

**Authors:** Rubén Tovar, Ana Luisa Gavito, Antonio Vargas, Laura Soverchia, Laura Hernandez-Folgado, Nadine Jagerovic, Elena Baixeras, Roberto Ciccocioppo, Fernando Rodríguez de Fonseca, Juan Decara

**Affiliations:** 1Instituto de Investigación Biomédica de Málaga (IBIMA), Hospital Universitario Regional de Málaga, UGC Salud Mental, Avda. Carlos Haya 82, Pabellón de Gobierno, 29010 Málaga, Spain; rubentovar7@hotmail.com (R.T.); analugavito@hotmail.com (A.L.G.); antoniovargasfuentes@gmail.com (A.V.); 2Facultad de Medicina, Universidad de Málaga, 29010 Málaga, Spain; 3Pharmacology Unit, School of Pharmacy, University of Camerino, Via Madonna delle Carceri 9, 62032 Camerino, Italy; laura.soverchia@unicam.it (L.S.); roberto.ciccocioppo@unicam.it (R.C.); 4Instituto de Química Médica, CSIC, C/Juan de la Cierva 3, 28006 Madrid, Spain; lhernandez@iqm.csic.es (L.H.-F.); nadine@iqm.csic.es (N.J.); 5Departamento de Bioquímica y Biología Molecular, Facultad de Medicina, Universidad de Málaga, 29010 Málaga, Spain; ebaixeras@uma.es

**Keywords:** N-acylethanolamine, palmitoleylethanolamide, monounsaturated fatty acids, palmitoleic acid, oleylethanolamide, oleic acid, peroxisome proliferator receptor alpha, cannabinoid receptors type 1, hypercaloric cafeteria diet

## Abstract

Obesity is currently a major epidemic in the developed world. However, we lack a wide range of effective pharmacological treatments and therapies against obesity, and those approved are not devoid of adverse effects. Dietary components such as palmitoleic acid have been proposed to improve metabolic disbalance in obesity, although the mechanisms involved are not well understood. Both palmitoleic acid (POA) and oleic acid (OA) can be transformed in N-acylethanolamines (NAEs), mediating the effects of dietary POA and OA. To test this hypothesis, here, we study the effects on food intake and body weight gain of palmitoleylethanolamide (POEA) and the OA-derived NAE analogue, oleoylethanolamide (OEA), in Sprague–Dawley rats with a hypercaloric cafeteria diet (HFD). Plasma biochemical metabolites, inflammatory mediators, and lipogenesis-associated liver protein expression were also measured. The results indicate that POEA is able to improve health status in diet-induced obesity, decreasing weight, liver steatosis, inflammation, and dyslipemia. The action of POEA was found to be almost identical to that of OEA, which is an activator of the nuclear peroxisome proliferator receptor alpha (PPARα), and it is structurally related to POEA. These results suggest that the dietary administration of either POA or POEA might be considered as nutritional intervention as complementary treatment for complicated obesity in humans.

## 1. Introduction

Although obesity is a widespread epidemic in the developed world, its pharmacological treatment is far from being optimized. Many assayed therapies against obesity have been retired due to cardiovascular or central adverse effects, where the few that remain are not totally effective, being able to produce unwanted side effects, limiting their acceptance. For this reason, a continuous research effort has been dedicated to identifying new therapies and strategies toward the development of new anti-obesity drugs [1]. The failure of classical targets against obesity, all of them focusing on the central nervous system—that is, serotonin receptor-acting compounds or antagonists of central cannabinoid receptors (CRs)—has deflected interest toward peripheral targets with the potential to control both appetite and energy expenditure without generating unwanted side effects. This challenge involves not only the identification of new chemical structures but also the validation of their biological targets for the therapeutic treatment of obesity. In this sense, the use of dietary components that might facilitate the activation of homeostatic mechanisms regulating appetite and metabolism provides further opportunities that deserve further exploration.

Among the various dietary components, monounsaturated fatty acids (MUFAs), such as oleic acid (OA) and palmitoleic acid (POA), have gained much attention, as nutritional analyses have revealed that they can produce beneficial effects on metabolic health. In addition, they can serve as substrates for the synthesis of a class of interesting endogenous lipid mediators—the N-acylethanolamides (NAEs)—in vitro. Current research has revealed that the NAE oleoylethanolamide (OEA) serves as a bioactive signal for the control of appetite, lipid metabolism, and inflammation. Interestingly, the physiological relevance of other NAEs, such as palmitoleylethanolamide (POEA), is practically unknown. NAEs can be found in several vegetable sources, especially seeds, such as peanuts, alfalfa, peas, beans, black-eyed peas, soya, and so on, indicating that they are nutritional components worth further in-depth study [2].

NAEs are a family of lipids capable of activating several lipid receptors with differential roles in controlling appetite and metabolism, including the endogenous cannabinoid receptors type 1 (CB1) and type 2 (CB2) [3] and the nuclear peroxisome proliferator receptor alpha (PPAR-α) [4]. Both CB1 and CB2 receptors belong to the so-called endocannabinoid system (ECS), which are main endogenous ligands; the ECS also involves N-arachidonoylethanolamine (AEA) [5], which is also called anandamide, and 2-arachidonoyl-glycerol (2-AG) [6], as well as the enzymes that catalyze endocannabinoid synthesis, uptake, and their metabolism [7]. CB1 antagonists have been assayed effectively for obesity, but unwanted affective disorders induced by its administration motivated its market withdrawal [8]; however, the discovery of the pro-lipogenic role of the ECS has motivated active research toward an understanding of its physiological importance in the development of obesity, as well as a search for additional pharmacological targets/interventions capable of modulating its activity [9].

As a counterpart to the actions of anandamide, OEA exerts opposite effects, promoting satiety [10], a reduction of lipogenesis [11], and analgesic and anti-inflammatory actions through the activation of peroxisome proliferator-activated receptor alpha (PPAR-α) [12], an additional molecular target for NAEs [13,14]. PPAR-α receptors, which belong to the ligand-activated transcription factors nuclear receptor superfamily (including steroid, glucocorticoid, and thyroid receptors), are involved in the regulation of appetite, food intake, and lipid metabolism, being an important regulator of energy homeostasis and metabolic function. PPAR-α is mostly present in the liver, kidney, heart, small intestine, and muscle, and it is targeted by fibrate hypolipidemic drugs, which is a class implicated in the catabolism of fatty acids, through increasing their oxidation [15]. Therefore, its agonists could play a significant role in the treatment of dyslipidemia or metabolic syndromes, by decreasing triglyceride levels in plasma [16], thus playing a significant role in glucose homeostasis and insulin resistance [17]. OEA is capable of interacting with peripheral sensory terminals in the gut, which are capable of regulating appetite through the activation of satiety signals acting directly in the sensory vagal afferent neurons that project to the paraventricular nucleus of the hypothalamus [18,19]. Taking into consideration the above-described roles of NAEs, it is important to highlight that they are produced from phospholipid precursors, such as N-acylphosphatidyl ethanolamines (NAPEs), which are generated by a specific N-acyltransferase (NAT) from available free fatty acids (Figure 1).

Oleic acid is the most common fatty acid in the human diet, and it is present in a great variety of vegetable oils, such as olive, pecan, canola, peanut, macadamia, sunflower, grape seeds, sesame, and so on. It is the most abundant fatty acid in human adipose tissue [20]. In addition, diets rich in OA have been shown to have beneficial health effects, such as increasing levels of high-density lipoprotein (HDL) cholesterol and lowering levels of low-density lipoprotein (LDL) cholesterol, and reducing cardiovascular disease risk factors [21]. Dietary OA content can promote OEA synthesis, contributing to a reduction of the subsequent energy intake in humans [22]. Regarding POA, dietary sources include breast milk and a variety of animal fats, as well as vegetable and marine oils, being a common component of acylglycerides in human adipose tissue. Although it is converted into POEA in the body, few experimental approaches have been published to date. It could be postulated that POEA plays a role in nutrient availability, coordinating metabolic responses to diet, and glucose management [23]. It may also play a potential regulatory role as a major signaling lipid that improves hepatic lipid metabolism and enhances insulin sensitivity, but conclusive experimental evidence is lacking [24]. Supporting this view, the administration of POA to insulin-resistant animals has been shown to potentially alleviate hyperglycemia and hyperlipidemia [25].

In order to clarify the role of POA in controlling appetite, lipid metabolism, and inflammation, we designed an experimental approach involving obese animals (through a highly caloric palatable cafeteria diet), which were treated with either OEA or POEA. The experimental design allowed us to establish the potential utility of POEA (and, eventually, POA) as a complementary nutritional ingredient for treating complicated obesity.

## 2. Materials and Methods

### 2.1. Animals, Diets, and Experimental Groups

Feeding studies and experiments related to standard diet (STD) and diet-induced obesity by palatable hypercaloric cafeteria diet (HFD) were performed in 64 Male Sprague–Dawley rats (Charles River) weighing 260–360 g at the beginning of the experiments. Animals were housed in pairs under a 12 h light/dark cycle (lights off at 21:00) in a room with controlled temperature (20–22 °C) and humidity (45–55%). Unless otherwise indicated, rats were fed water and standard rat chow pellets (STD; 4RF18, Mucedola, Settimo Milanese, Italy) until starting the diet-induced obesity procedure. All experiments were performed in accordance with the European directive 2010/63/EU governing animal welfare and protection, which is acknowledged by Italian Legislative Decree n. 116 (27 January 1992). Obesity-induced rats were randomly divided into two groups with comparable mean body weight (no significant difference). The first group (*n* = 32) was fed with standard laboratory chow ad libitum (2.6 kcal/g); the second group (*n* = 32) was fed ad libitum with a palatable HFD daily prepared diet (5.3 kcal/g). See Table 1 for food components and nutritional facts.

Diets were maintained for 13 weeks, until rats fed with HFD showed obesity and/or insulin resistance. From that point on, each group was subdivided in the respective experimental groups for vehicle (VEH), OEA, and POEA treatments for both STD and HFD. Figure 2 shows the intake scheme and the experimental groups used in this study. Body weight and food intake for all groups were registered daily during the study (Figure 3).

### 2.2. Glucose Tolerance Test

To determine the effectiveness of diet on the induction of insulin resistance, we performed a glucose tolerance test. For the glucose tolerance test (GTT), both STD and HFD rats were food deprived for 18 h before the procedure. Animals were administered an intraperitoneal (i.p.) glucose overload of 2 g/kg body weight. Tail blood samples were collected before and 0, 5, 10, 15, 30, 60, and 120 min after glucose administration. Glucose was determined using a standard glucose oxidase method, as described previously [26].

### 2.3. Drugs and Chronic Treatment

OEA and POEA were synthesized at the Instituto de Química Médica (Madrid, Spain), as previously described [11]. The NAEs were dissolved in a vehicle composed of 5% Tween 80 (Sigma) diluted 95% (vol/vol) in a 0.9% saline solution. The drugs were injected i.p. at a dose of 10 mg/kg of body weight in a volume of 1 mL/kg of body weight for 15 days. The doses were selected based on previous experimental procedures for OEA [10].

### 2.4. Sample Collection

At the end of chronic treatment, animals of the two experimental groups were euthanized (by decapitation) two hours after the last administration. Blood samples were collected (EDTA-2Na tubes) and centrifuged (1000× *g* for 10 min at 4 °C). Plasma samples were frozen at −80 °C for biochemical and hormonal analyses. Livers were dissected and frozen at −80 °C until protein expression analyses.

### 2.5. Measurement of Metabolites, Hepatic Enzymes, and Hormones in Plasma

The following metabolites were measured in plasma: glucose, triglycerides, total cholesterol, high-density lipoprotein (HDL), bilirubin, urea, uric acid, creatinine, and the hepatic enzymes alanine aminotransferase (ALT) and aspartate aminotransferase (AST). They were analyzed using commercial kits, according to the manufacturer’s instructions, in a Hitachi 737 Automatic Analyzer (Hitachi, Tokyo, Japan). Very-low-density lipoprotein (VLDL) and low-density lipoprotein (LDL) were determined through use of modified Friedewald equations: VLDL = triglycerides/5; LDL = cholesterol – ((triglycerides/5) + HDL) [27]. The plasma levels of insulin and interleukin 6 (IL-6) were measured using a commercial rat insulin enzyme-linked immunosorbent assay (ELISA) kit (EZRMI-13K Millipore, Missouri, USA), a Rat IL-6 ELISA Kit (KRC006 Novex, by Life technologies, USA), and a tumor necrosis factor alpha (TNF-α) Kit (KRC3011 Novex, by Life technologies, USA). Homeostatic Model Assessment for Insulin Resistance (HOMA IR) was calculated using the following formula: HOMA IR = (fasting plasma insulin (uIU/mL) × fasting glycemia (mmol/L)/22.5) [28].

### 2.6. Hepatic Lipid Extraction and Fat Content

Fat extraction was performed as previously described [29]. Briefly, total lipids from frozen liver samples were extracted with chloroform–methanol (2:1, *v*/*v*) and butylated hydroxytoluene (0.025%, *w*/*v*), according to the Bligh and Dyer method. The lower phase containing lipids was separated after two centrifugation steps (2800× *g*, 4 °C for 10 min) and dried by nitrogen. The liver fat content was expressed as a percentage of the tissue weight.

Liver samples were analyzed for fat deposits by oil red O staining. Frozen samples were cut into 30 µm thick sections using a sliding microtome (Leica SM200R, Wetzlar, Germany) and fixed with 10% formal calcium. Sections were washed with distilled water and rinsed with 60% isopropanol. Then, the sections were stained with freshly prepared oil red O (Sigma, St. Louis, MO, USA) working solution for 20 min (oil red O stock stain: 0.5% of oil red O in isopropanol; oil red working solution: 30 mL of the stock stain and 20 mL of distilled water). Sections were rinsed with 60% isopropanol, counterstained with Mayer’s hematoxylin, rinsed with tap water, and mounted in aqueous media. Images for each section were acquired with a DP70 digital camera (Olympus Iberia, S.A., Barcelona, Spain) connected to an Olympus BX41 microscope with 40x objective lens. Five to eight rats per group and two section of liver per animal were used. Then, red stain intensity was quantified by densitometry, using the image processing software ImageJ (Rasband, W.S., ImageJ, U.S., NIH; http://imagej.nih.gov/ij, accessed on 20 May 2021).

### 2.7. Protein Extraction and Western Blot Analysis

Western blotting was performed as described previously [30]. Proteins from tissue portions of liver were extracted on ice using a lysis buffer (40 mM Tris-HCl, pH 7.4; 200 mM NaCl; 4% Triton ×100; 20 mM EDTA, pH 8) containing a proteinase and phosphatase inhibitor cocktail (5 mg/mL leupeptin, 100 mM NaF, 1 mM sodium orthovanadate, 5 μg/mL aprotinin, 1 μg/mL pepstatin A, 10 μg/mL trypsin inhibitor, 1 mM phenylmethylsulfonyl fluoride, 0.75 μL/mL protease, and phosphatase inhibitor cocktail). Protein concentration was determined by Bradford protein assay. Equivalent amounts of protein extract (30 μg) were separated in 10% SDS-PAGE gels (NuPAGE™ Novex™ 10% Bis-Tris Midi Protein Gels, #WG1201A; Thermo Fisher, Waltham, MA, USA) using an SDS running buffer (NuPAGE^®^ MOPS SDS Running Buffer, #NP0001; Thermo Fisher, Waltham, MA, USA); then, they were electroblotted onto 0.2 μm nitrocellulose membranes (Bio-Rad). Membranes were blocked in TBS-T (50 mM Tris-HCl, pH 7.6; 200 mM NaCl; and 0.1% Tween 20) with 2% albumin fraction V from bovine serum (BSA; Roche) for 1 h at room temperature. Specific proteins were detected by incubation overnight at 4 °C and raised against the following rabbit antibodies: fatty acid synthase (FAS) (#3180S; Cell Signalling Technology, Danvers, MA, USA), fatty acid amide hydrolase (FAAH) (#101600; Cayman, Ann Arbor, MI, USA), and stearoyl-CoA desaturase (SCD1) (#ab19862; Abcam, Cambridge, UK). The mouse anti-adaptin-γ (#610385; BD Biosciences, Madrid, Spain) was used as a reference protein. All antibodies were diluted 1:1000 in Tris-buffered saline containing 1% Tween (TBS-T) and 2% bovine serum albumin (BSA). After extensive washing in TBS-T, HRP-conjugated anti-rabbi or anti-mouse IgG (H+L) secondary antibodies (Promega, Madison, WI, USA) diluted 1:10,000 were added for 1 h at room temperature. After enhanced chemiluminescence detection (Santa Cruz Biotechnologies, Dallas, TX, USA) in an Autochemi-UVP Bioimaging System, bands were quantified by the ImageJ software.

### 2.8. Statistical Analysis

Data are expressed as the mean ± standard error of the mean (SEM) of at least eight determinations per experimental group. Statistical results were obtained using the computer program GraphPad Prism version 6.01 (GraphPad Software Inc., San Diego, CA, USA). Differences were analyzed by one- and two-way ANOVA, depending on the factors and kind of analyses, followed by Bonferroni post hoc tests for multiple comparisons or Student’s unpaired *t*-tests, when appropriate. A *p*-value below 0.05 was considered statistically significant.

## 3. Results

### 3.1. Comparative Effects of OEA and POEA on Body Weight in STD and HFD

Exposure to palatable HFD rapidly induced obesity, as revealed by body weight acquisition. The separation of the weight gain curves regarding diet use was significant after two weeks of exposure (Figure 3A), while the cumulative kcal intake increased from the first day of the diet (Figure 3B). Finally, after 91 days—when both diet groups were totally differentiated—the rats of each group were further separated into three groups for chronic treatment purposes: vehicle (VEH), OEA, and POEA. In addition, as expected, the GTT showed that the animals fed with a HFD presented insulin resistance (Appendix A in Appendix A). Daily NAEs administration at a dose of 10 mg/kg for 15 days resulted in a significant reduction of body weight in both STD and HFD groups. In the case of the STD group, OEA and POEA showed a significant reduced body weight, starting from day 13 (*p <* 0.01 and *p <* 0.001, respectively; Figure 4A,B). The treatments with both OEA and POEA in HFD were more effective than in STD, showing differences from day 8 of treatment (*p <* 0.05), which increased in the following days (*p <* 0.01 and *p <* 0.001, respectively; Figure 4C,D). Then, at the end of the study, the treatment with NAEs in the STD group resulted in a significant decrease in body weight by at least 10 g (*p <* 0.05; Figure 4E) while being more than 20 g for NAEs in HFD (*p <* 0.001; Figure 4F). These data indicate the effectiveness of NAEs in the reduction of body weight when obesity is induced by a palatable HFD.

### 3.2. OEA and POEA Reduced Food Intake in Diet-Induced Obese Rats

The chronic administration of NAEs in STD did not show differences in total food intake or cumulative kcal intake along the 15 days of treatment (Figure 5A,C). In contrast, the same treatment in HFD was able to decrease both total food and kcal intake. Both OEA and POEA were equally effective in reducing feeding (*p <* 0.05; Figure 5B,D).

### 3.3. Effects of OEA and POEA on the Plasma Biochemistry in STD-Fed Animals

Biochemical analysis indicated that neither OEA nor POEA affected the plasma levels of glucose, triglycerides, high-density lipoprotein (HDL), low-density lipoprotein (LDL), very low-density lipoprotein (VLDL), urea, uric acid, creatinine, bilirubin, and transaminases such as aspartate transaminase (AST) and alanine transaminase (ALT) in animals fed STD. However, we observed a reduction in the plasma levels of triglycerides (*p <* 0.01), cholesterol (*p <* 0.01), and HDL (*p <* 0.001) after OEA treatment in STD, with respect to VEH (see Table 2). POEA only reduced triglycerides (*p <* 0.01). Other biochemical parameters or metabolites related to hepatic and renal functions were not affected, indicating a lack of toxicity after treatment with either POEA or OEA.

### 3.4. Effects of OEA and POEA on the Plasma Biochemistry in HFD-Fed Animals

The ANOVA analysis showed that HFD results in a metabolic disruption reflected in the dyslipidaemia, hyperglycemia, hyperuricemia, and uremia observed in comparison to STD-fed animals (see Table 2 and Table 3).

Both treatments, with either OEA or POEA, induced a significant restoration of the metabolic equilibrium, as reflected by the reduction of multiple parameters. Thus, glucose levels were reduced (*p <* 0.001); see Table 3. Similarly, both OEA and POEA had a significant main effect on the total triglycerides level (*p <* 0.05 and *p <* 0.01, respectively) in HFD. The total plasma cholesterol was not affected, but HDL with OEA (*p <* 0.05), LDL, and VLDL with both OEA (*p <* 0.05) and POEA (*p <* 0.05) showed a reduced level compared with the VEH group.

When the effects of both diets were compared, HFD produced significant negative effects on all biochemical parameters evaluated in plasma, such as uric acid, urea, creatinine, and bilirubin, as well as markers of liver damage (AST and ALT). The results, analyzed by two-way ANOVA, showed that both the OEA and POEA treatments were also capable of normalizing plasma biochemical parameters related to liver or kidney toxicity. Compared with VEH, OEA showed a marked improvement in uric acid (*p <* 0.001), urea (*p <* 0.01), creatinine (*p <* 0.01), and bilirubin (*p <* 0.05) levels in plasma. Similar effects were observed with POEA, in terms of uric acid (*p <* 0.01), urea (*p <* 0.05), creatinine (*p <* 0.05), and bilirubin (*p <* 0.05) levels.

The positive effects of both NAEs can be also detected in AST, the level of which both OEA and POEA reduced in the plasma (*p <* 0.05 and *p <* 0.01, respectively). In the case of ALT, only OEA was able to improve its level in plasma (*p <* 0.01).

### 3.5. Effect of OEA and POEA on Insulin Resistance

As exposure to HFD results in insulin resistance and diabetes, we investigated the effect of chronic treatment with both NAEs on circulating insulin concentrations and insulin resistance. In the HFD animals treated with VEH, plasma insulin levels were significantly increased when compared to the STD VEH group. OEA treatment decreased circulating insulin in both the HFD and STD groups (*p <* 0.05), while POEA only reduced insulin in HFD-fed animals (Figure 6A). HOMA IR, as a method for assessing β-cell function and insulin resistance (IR), was measured after chronic NAEs treatments in both STD and HFD. OEA was able to decrease the HOMA IR index in both diets (*p <* 0.001). The positive effect of OEA on IR was also observed after POEA administration, although only in HFD (Figure 6B).

### 3.6. NAEs Reduce the Plasma Concentration of Pro-Inflammatory Cytokines IL-6 and TNF-α

A hallmark of HFD-induced obesity is inflammation. Thus, we explored the impact of NAE treatment on the plasma levels of the pro-inflammatory cytokines IL-6 and TNF-α. As described, important differences were observed, with respect to circulating concentrations of both IL-6 and TNF-α when STD and HFD animals treated with VEH were compared. The concentrations of both cytokines were increased as a result of HDF-induced obesity, but the effect was counteracted by both OEA and POEA treatments (Figure 7A,B). In STD-fed animals, only the level of TNF-α was reduced by both POEA and OEA.

### 3.7. Effects of Repeated OEA and POE Administration on the Expression of Enzymes Involved in Fatty Acid Synthesis and NAEs Degradation in the Liver

Analysis of protein expression in livers showed that both OEA and POEA are capable of modulating the expression of lipogenic enzymes FAS and SCD1 in STD-fed animals. While OEA decreases the expression of both enzymes, POEA only decreases SCD1 levels (Figure 8A,B). FAAH, an enzyme that degrades NAEs, significantly reduced the liver expression of POEA, suggesting that it facilitates the half-life of NAEs in the liver (*p <* 0.05; Figure 8C).

Regarding HFD-fed animals, analysis of protein expression levels showed no alterations in FAS after OEA or POEA administration (Figure 9A). The SCD1 enzyme level presented a more marked decrease after OEA treatment (*p <* 0.01) than STD (Figure 9B), but it not show differences in other NAEs. Contrary to STD, the enzyme of cannabinoid degradations, FAAH, showed a significant decrease only after OEA treatment (*p <* 0.05; Figure 9C).

### 3.8. Analysis of Liver Steatosis after NAEs Treatment

After chronic NAE treatment, the total lipids and their content were analyzed in livers by both oil red O stain quantification and organic extraction. In the analyzed images, the STD-fed group displayed less fat droplet content (Figure 10A) than the HFD group, which is compatible with severe steatosis (Figure 10B,C). Treatment with both OEA and POEA reduced the severe steatosis associated with HFD, as well as a decrease in the total lipid content in livers of STD-fed animals (*p <* 0.01) when compared with the respective vehicle groups (Figure 10D–G). These results were confirmed by the quantification of fat by organic extraction, with OEA proving to be more potent than POEA (Figure 10H).

## 4. Discussion

The presented results demonstrate that the natural OEA analogue, POEA, is capable of reducing food intake and body weight gain as well as correcting both the metabolic changes and liver steatosis induced by exposure to a palatable high fat content cafeteria diet. It is especially remarkable how both compounds are capable of reducing insulin resistance by lowering glycemia and plasma insulin concentrations as well as obesity-associated inflammatory markers such as IL-6 and TNF-α. Although the experimental approach only used one dose of both OEA and POEA, the data suggest that both compounds were almost equal in the normalization of the diet-induced obesity metabolic disruptions, although OEA was apparently more efficient in reducing liver fat content or inhibiting the lipogenic enzyme FAS. The current approach used the parenteral route to assess the pharmacological activity of both compounds. OEA has been demonstrated to have feeding reduction activity when given by the oral route, in both formulated and non-formulated administrations [31]. We lack information on POEA activity through the oral route; however, it is reasonable to think that it will be active when administered orally. This is a remarkable question that needs to be solved in the case of potential use of POEA as an active nutritional ingredient for complicated obesity. In any case, it is important to highlight that POEA is one of the less-abundant NAEs found in seeds and processed foods when compared with the most abundant ones, such as linoleoylethanolamide (LEA), OEA, or palmitoylethanolamide (PEA) [32]. Thus, its potential use as a food ingredient might be considered under the form of a food additive.

Our observations indicate that the chronic administration of OEA and POEA induced body weight loss in rats under both STD and HFD feeding conditions. Interestingly, this slimming effect was more pronounced in HFD-fed rats, especially in the group treated with OEA. Furthermore, it was associated with a slight but significant anorectic effect of both NAEs treatment in HFD-fed rats but not in STD-fed rats. Of note is that both the slimming and anorectic effects of OEA and POEA were also paralleled with a significant improvement in plasma biochemical parameters, with respect to the HFD-fed control group. Moreover, these biochemical parameters recovered levels very close to those found in STD-fed and vehicle-treated rats. This effect could be due, in part, to the reduced food intake and body mass observed in the HFD-fed group. Even so, the tendency of re-establishing the plasma levels of the parameters evaluated was so robust, it strongly indicated that NAEs treatment exerts a specific protective action on the integrity of liver and kidney tissues in obese rats. Indeed, the almost-normal levels of bilirubin, ALT, HDL, creatinine, uric acid, and so on, support this notion.

Regarding the mechanism of action of POEA, we did not find any published data on the efficacy of POEA in terms of the PPAR-α receptor. However, as both OEA and POEA display an almost identical chemical structure (Figure 1), it is feasible to think that POEA might also act as an agonist on the nuclear receptor that regulates food intake, lipid metabolism, and inflammation [11]. POEA differs only in the length of the acyl chain, which is reduced by two carbons (Figure 1), and it has the same polar head as OEA. In addition, two important findings suggest that they share similar PPAR-α-dependent mechanisms of action: they both reduce the circulating concentration of triglycerides—a hallmark of marketed hypolipemiants that are activators of the PPAR-α, such as the fibrates [33]—and they efficiently reduce the lipogenic enzyme SCD1, whose action also depends on the PPAR-α receptor [34]. In addition, the reduction of both IL-6 and TNF-α is a well-known PPAR-α receptor-dependent action, which counteracts nuclear factor kappa-light-chain-enhancer of activated B cells (NFκB)-dependent inflammation, and which has been proposed to mediate NAEs-induced anti-inflammatory actions on intestinal inflammation [35,36]. This anti-inflammatory action of OEA has been also demonstrated in the brain after lipopolysaccharides (LPS) administration [37], further supporting the important role of PPAR-α receptor-activating NAEs. NAPEs are cleaved by the stimulus-dependent activation of NAPE-specific phospholipase D (NAPE-PLD), releasing the corresponding NAE. Thus, nutritional supplements including specific fatty acids might lead to an increased availability of specific NAPEs, which act as precursors for NAEs [38]. Therefore, it might be possible that the described positive effects of both OA and POA MUFAs in the diet (healthy blood lipid profiles, low blood pressure, improvement of insulin sensitivity, and regulation of glucose levels) might derive from their biotransformation into NAEs, especially those that activate PPAR-α receptors, such as OEA. [39].

Additional described targets for OEA are the receptors G protein-coupled receptor 55 (GPR55) and the G protein-coupled receptor 119 (GPR119), both of which are involved in the regulation of insulin secretion and appetite [40,41]. We did not find any data on the interaction of POEA at GPR55, but a recent work has described that POEA has a very similar profile to OEA on GPR119, being able to stimulate GPR119-dependent cyclic adenosine monophosphate (cAMP) production and the GPR119-dependent release of glucagon-like peptide-1 (GLP-1) from neuroendocrine cells [24]. Thus, in addition to PPAR-α-dependent actions, it is reasonable to believe that POEA might improve insulin resistance and reduce feeding by modulating pancreatic islet physiology and appetite, acting through the GPR119 receptor.

POEA derives from palmitoleic acid (POA), a monounsaturated, omega-7 fatty acid present in some vegetable seeds (i.e., macadamia nuts), fish oil, and breast milk. POA can be also synthesized in vitro by the action of delta-9 desaturase (often called SCD1), which catalyzes the desaturation of palmitic acid. Synthesized POA is incorporated into triglycerides and cell membrane phospholipids, accounting for up to 3.2% of total fatty acids [42]. It is notable that POA is an activator of the PPAR-α receptor, and its use as an active food ingredient has been proposed for the treatment of complicated obesity. The basis for this nutritional intervention was the finding that POA can be released from adipocytes, acting as a lypokine that enhances glucose disposal and attenuates liver steatosis in diet-induced obesity [25]. The administration of POA resulted in attenuated liver inflammation in a non-alcoholic fatty liver model and reduced atherosclerosis in human and pre-clinical models. These are exactly the same results that we found in our analysis of the action of POEA, including a reduction of liver inflammation and damage. These striking similarities suggest that there exists a potential link between these compounds. One possibility is that POEA may be converted into POA by the action of FAAH. This enzyme, which degrades NAEs, was not affected after POEA administration (Figure 9) and, as such, this is a plausible hypothesis that deserves further analysis. POA released from the intracellular degradation of POEA might act on PPAR-α receptors to promote the beneficial changes observed in the present study. Another possibility is that POEA itself can directly activate PPAR-α receptors in a similar way as OEA, in order to promote a healthy adaptive response to an HFD. In both cases, whether POEA is a precursor of POA or a direct activator of PPAR-α receptors, its use as a healthy food ingredient was demonstrated. Further analysis is needed in order to understand the pharmacokinetics of this bioactive lipid, its final mechanism of action, and the optimal formulation for its administration. Previous studies in our laboratory with nanoemulsions as an optimal vehicle for the oral administration of OEA might provide clues to find the best way of administering this bioactive lipid, which can serve as a simple nutritional intervention to fight complicated obesity.

## 5. Conclusions

POEA is a palmitoleic acid-derived NAE, whose parenteral administration resulted in a reduction in body weight gain, food intake, liver steatosis, dyslipidemia, insulin resistance, and inflammation associated with exposure to a high-fat palatable diet in the conducted experiment. Its pharmacological activity resembles that of the oleic acid derivative OEA, which is a well-characterized PPAR-α/GPR119 receptor agonist that is capable of counteracting the metabolic syndrome associated with complicated obesity. Whether POEA is a precursor of the bioactive POA or a direct PPAR-α receptor agonist remains to be elucidated, but its bioactivity indicates its potential use as a healthy nutritional ingredient in humans suffering metabolic disturbances associated with diet-induced obesity.

## Figures and Tables

**Figure 1 nutrients-13-02589-f001:**
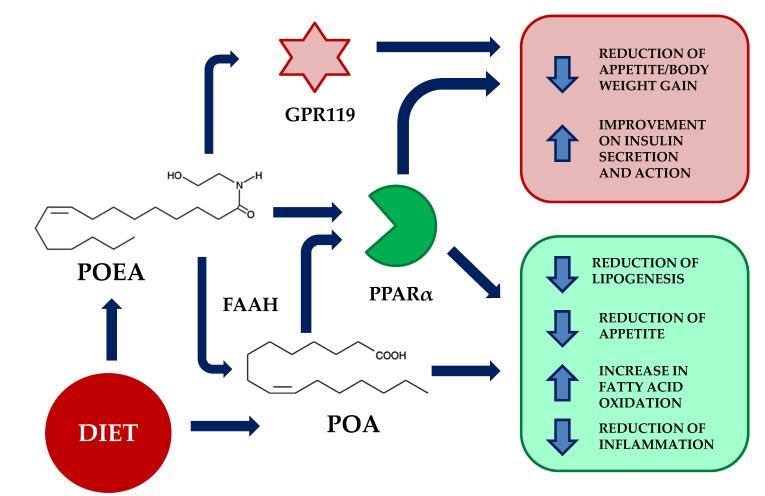
Potential mechanisms of action of palmitoleylethanolamide (POEA). POEA is the N-acylethanolamine of palmitoleic acid (POA). Upon dietetic administration, POEA might act directly through peroxisome proliferator-activated receptor alpha (PPAR-α) receptors, promoting the inhibition of lipogenesis, reduction of appetite, increase in fatty acid oxidation, and reduction of inflammation. These actions are also mimicked by palmitoleic acid, its main degradation product. In addition, POEA has been found to activate the orphan G protein-coupled receptor 119 (GPR119), resulting in a reduction of appetite and improvement of insulin action. FAAH; fatty acid amide hydrolase.

**Figure 2 nutrients-13-02589-f002:**
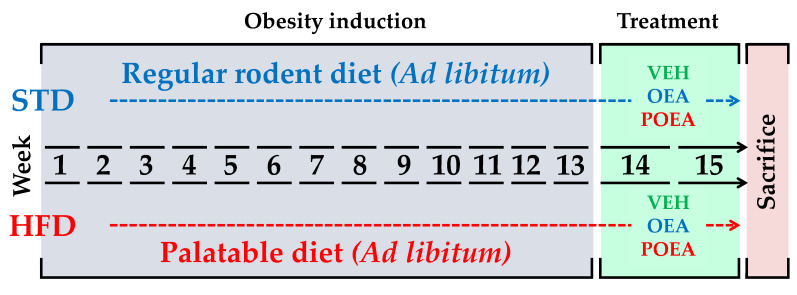
Experimental design. Animals exposed to either regular standard diet chow (STD) or a hypercaloric cafeteria diet (HFD) for 13 weeks were subsequently treated for two weeks with either vehicle (VEH), N-oleylethanolamine (OEA) and N-palmitoleoylethanolamine (POEA).

**Figure 3 nutrients-13-02589-f003:**
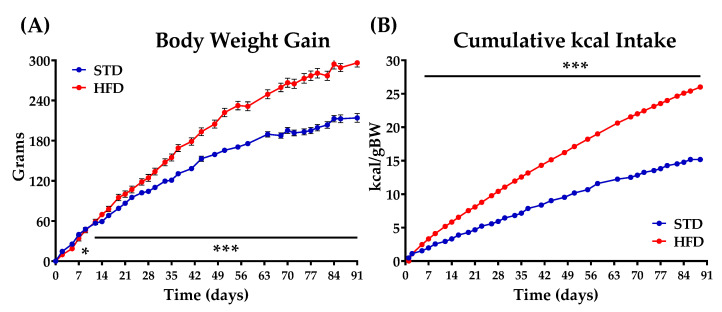
Body weight gain (**A**) and cumulative kcal intake (**B**) in animals fed for 91 days (13 weeks) with either a standard diet chow (STD) or a hypercaloric cafeteria diet (HFD). Values are expressed as mean ± standard error of the mean (SEM) (*n* = 32 animals/group). Two-way ANOVA and Bonferroni post hoc test: (*) *p* < 0.05 and (***) *p* < 0.001 vs. STD group.

**Figure 4 nutrients-13-02589-f004:**
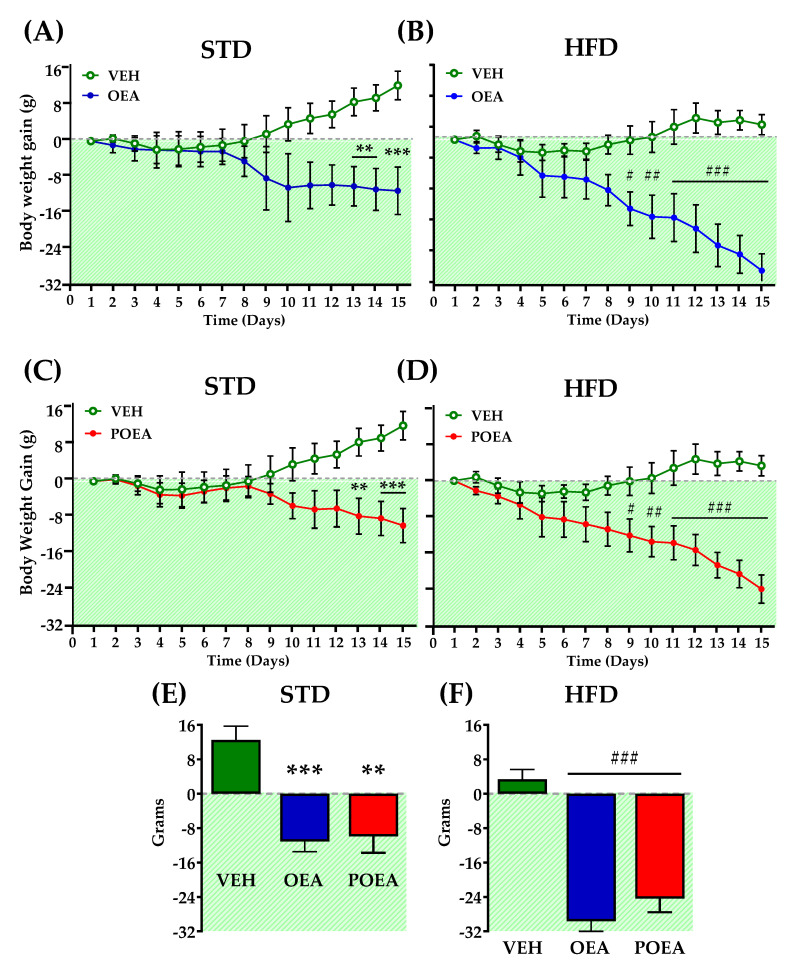
Effects of sub-chronic (15 days) administration of a 10 mg/kg dose of either oleoylethanolamide (OEA) (**A**,**B**) or palmitoleylethanolamide (POEA) (**C**,**D**) in animals fed with either a standard diet (STD) or hypercaloric cafeteria diet (HFD). Data are the evolution of cumulative body weight gain in male Sprague–Dawley rats. (Panels **E**,**F**) show the total weight gain variation after 15 days of treatment. Points or bars are means ± standard error of the mean (SEM) (*n* = 8 animals per group). Data were analyzed by two-way ANOVA (diet and time) and Bonferroni’s post hoc test. (**) *p* < 0.01, and (***) *p* < 0.001, significant differences compared with vehicle (VEH) STD group. (#) *p* < 0.05, (##) *p* < 0.01, and (###) *p* < 0.001 significant differences compared with VEH HFD group.

**Figure 5 nutrients-13-02589-f005:**
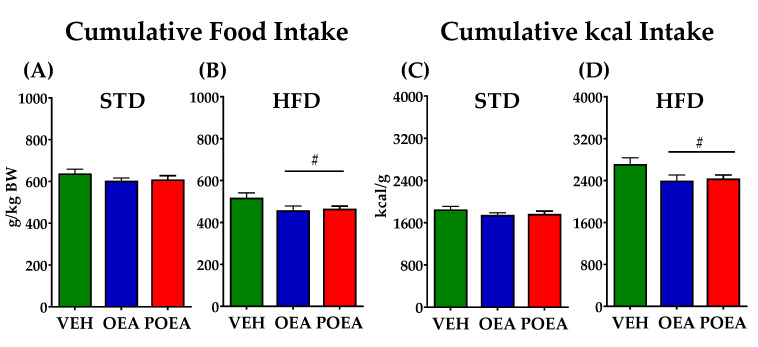
Effects of sub-chronic (15 days) administration of a 10 mg/kg dose of either oleoylethanolamide (OEA) or palmitoleylethanolamide (POEA) on food intake, which was measured as g/kg body weight (**A**,**B**) or cumulative kcal (**C**,**D**). Points or bars are means ± standard error of the mean (SEM) (*n* = 8 animals per group). Unpaired *t*-test: (#) *p <* 0.05 vs. vehicle (VEH) group.

**Figure 6 nutrients-13-02589-f006:**
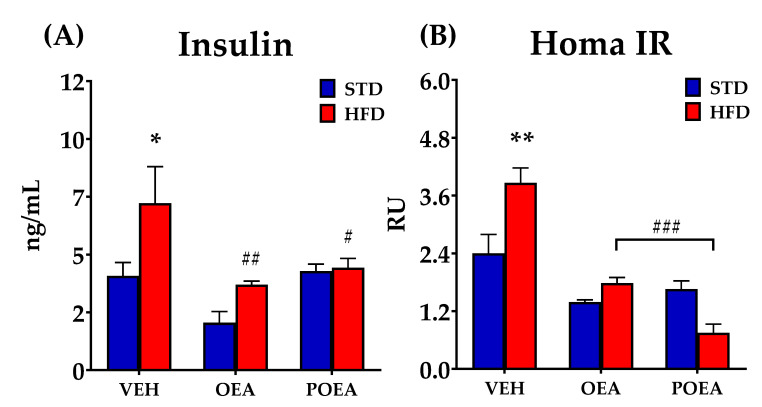
Effects of sub-chronic (15 days) administration of a 10 mg/kg dose of either oleoylethanolamide (OEA) or palmitoleylethanolamide (POEA). on the plasma levels of insulin (**A**) and analyzed the Homeostatic Model Assessment for Insulin Resistance (HOMA IR) (**B**) in male Sprague–Dawley rats. Data are expressed as means ± standard error of the mean (SEM) (*n* = 8 animals/group) analyzed by two-way ANOVA (diet and treatment and Bonferroni post hoc test). (*) *p <* 0.05 and (**) *p <* 0.01 significant differences compared with standard diet (STD) vehicle (VEH) group; (#) *p <* 0.05, (##) *p <* 0.01, and (###) *p <* 0.001 significant differences compared with hypercaloric cafeteria diet (HFD) VEH group.

**Figure 7 nutrients-13-02589-f007:**
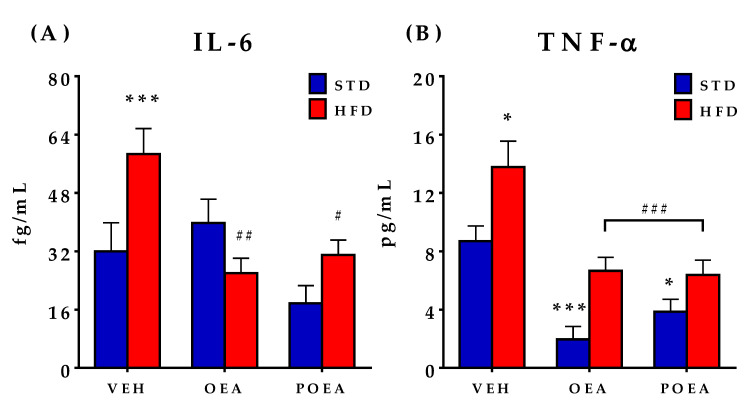
Effects of sub-chronic (15 days) administration of a 10 mg/kg dose of either oleoylethanolamide (OEA) or palmitoleylethanolamide (POEA) on the plasma levels of cytokines in male Sprague–Dawley rats: (**A**) Interleukin 6 (IL-6); and (**B**) tumor necrosis factor alpha (TNF-α), which were evaluated at the end of the treatment. Values are presented as means ± standard error of the mean (SEM) (*n* = 6–8 animals/group) analyzed by two-way ANOVA (diet and treatment and Bonferroni post hoc test). (*) *p <* 0.05 and (***) *p <* 0.001 significant differences compared with standard diet (STD) vehicle (VEH) group; (#) *p <* 0.05, (##) *p <* 0.01, and (###) *p <* 0.001 significant differences compared with hypercaloric cafeteria diet (HFD) VEH group.

**Figure 8 nutrients-13-02589-f008:**
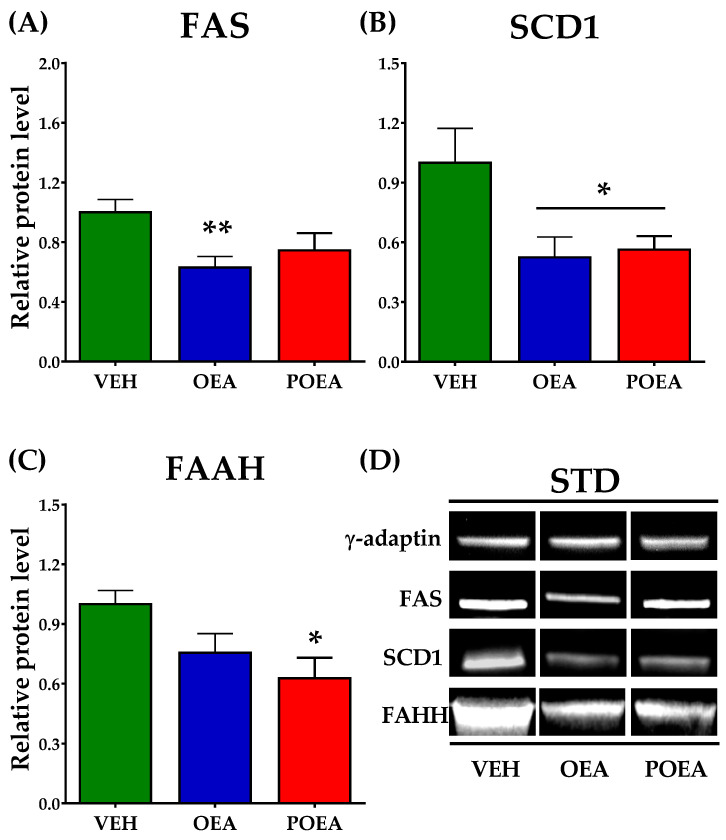
Analysis of liver protein expression of lipogenic enzymes and N-acylethanolamines (NAEs) degrading enzyme after sub-chronic (15 days) administration of a 10 mg/kg dose of either oleoylethanolamide (OEA) or palmitoleylethanolamide (POEA) in standard diet (STD)-fed Sprague–Dawley rats. Fatty acid synthase (FAS) (**A**), stearoyl-CoA desaturase (SCD1) (**B**), and fatty acid amide hydrolase (FAAH) (**C**) were analyzed by imaging densitometry. The bars show the results from six independent samples for each treatment group. (**D**) Representative images of Western blot analysis of liver samples for each treatment. The corresponding expression of γ-adaptin is showed as a loading control per lane. All samples were derived at the same time and processed in parallel. The bars were determined the means and corrected for γ-adaptin protein as reference ± standard error of the mean (SEM) (*n* = 6 samples per group). Unpaired *t*-test: (*) *p <* 0.05 and (**) *p <* 0.01 vs. vehicle (VEH) group.

**Figure 9 nutrients-13-02589-f009:**
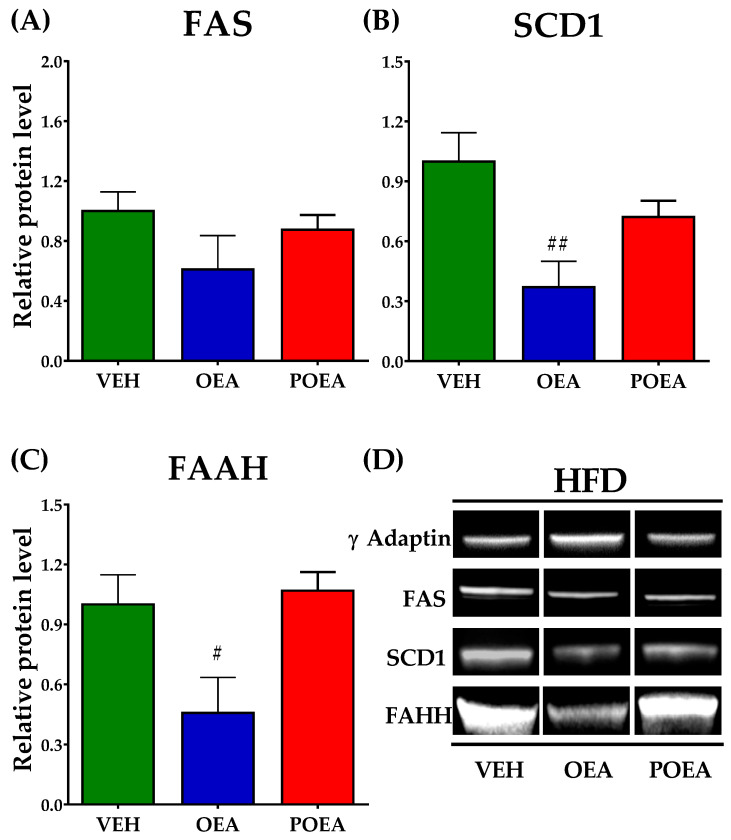
Analysis of protein expression after sub-chronic (15 days) administration of a 10 mg/kg dose of either N-acylethanolamines (NAEs) degrading enzyme after sub-chronic (15 days) administration of a 10 mg/kg dose of either oleoylethanolamide (OEA) or palmitoleylethanolamide (POEA) in liver in hypercaloric cafeteria diet (HFD)-fed Sprague–Dawley rats. Fatty acid synthase (FAS) (**A**), stearoyl-CoA desaturase (SCD1) (**B**), and fatty acid amide hydrolase (FAAH) (**C**) were analyzed by imaging densitometry. The bars show results from six independent samples from each treatment group. (**D**) Representative images of Western blot analysis of liver samples for each treatment. The corresponding expression of γ-adaptin is showed as a loading control per lane. All samples were derived at the same time and processed in parallel. The bars determined the means and corrected for γ-adaptin protein as reference ± standard error of the mean (SEM) (*n* = 6 samples per group). Unpaired *t*-test: (#) *p <* 0.05 and (##) *p <* 0.01 vs. vehicle (VEH) group.

**Figure 10 nutrients-13-02589-f010:**
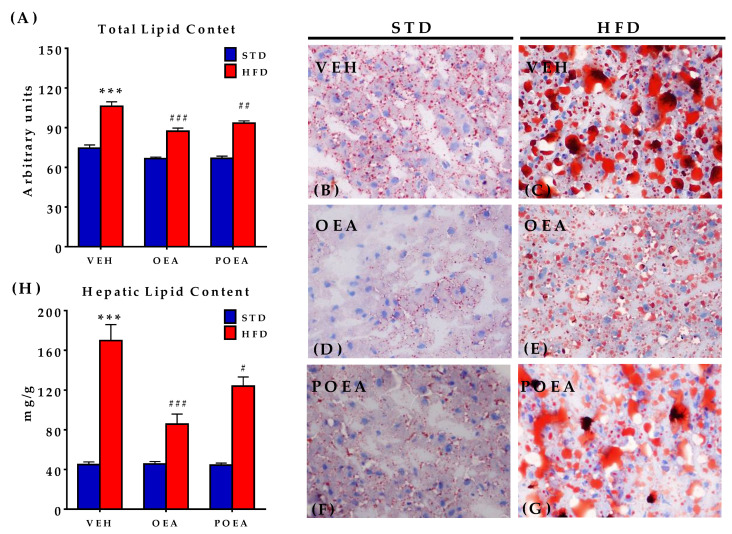
Effect of sub-chronic (15 days) administration of a 10 mg/kg dose of either oleoylethanolamide (OEA) or palmitoleylethanolamide (POEA) in liver in hypercaloric cafeteria diet (HFD) male Sprague–Dawley rats on hepatic lipid content (**A**) in standard diet (STD) and HFD, and corresponding representative images of liver fat content with oil red O staining, taken under a microscope with 40× objective (**B–G**). (**H**) Measurement of the total fat content in liver. Bars are means ± standard error of the mean SEM per group (*n* = 6 samples per group) analyzed by two-way ANOVA (diet and treatment, with Bonferroni post-hoc test). (***) *p* < 0.001 significant differences compared with STD vehicle (VEH) group; (#) *p <* 0.05, (##) *p <* 0.01, and (###) *p <* 0.001 significant differences compared with HFD VEH group.

**Table 1 nutrients-13-02589-t001:** Standard diet (STD) description and detail of the palatable hypercaloric cafeteria diet (HFD) products administered to obesity-induced animals.

STD	Regular Rodent Diet	Amount (%)	kcal (g)	FAT per 100 g of food	
				Saturated	Unsaturated
Rat chow pellets	(4RF18) Mucedola	100	2.60	1.4	5.0
Total kcal/g of STD Food				2.60	
Palatable HFD	Description (Italian Food)	Amount (%)	kcal (g)	FAT per 100 g of food	
				Saturated	Unsaturated
Mortadella	*Mortadella Italiana*	10	3.21	8.0	17.0
Rusk	*Crostini*	8	5.53	5.4	6.6
Cookie with cream	*Macine con panna*	18	5.44	5.1	12.9
Muffin with chocolate	*Muffin con ciocolatto*	22	4.56	3.1	13.9
Cheese	*Parmeggiano Reggiano*	14	4.38	17.0	11.0
Snack	*Patatine al formaggio*	16	5.35	3.3	21.7
Flavored lard	*Lardo*	12	9.04	34.0	53.0
Total kcal/g of HFD Food				5.30	

**Table 2 nutrients-13-02589-t002:** Plasma biochemical parameters from animals fed STD and treated with either OEA or POEA (10 mg/kg) for 15 days.

Standard Diet (STD)			
Plasma Metabolites	VEH	OEA	POEA
Glucose (mg/dL)	164.00 ± 8.45	168.57 ± 4.50	160.33 ± 12.37
Triglycerides (mg/dL)	150.29 ± 12.27	161.20 ± 8.82	136.33 ± 7.93
Cholesterol (mg/dL)	68.83 ± 4.80	48.29 ± 3.76 *	69.83 ± 7.42
VLDL (mg/dL)	27.67 ± 1.30	30.06 ± 2.45	32.24 ± 1.76
LDL (mg/dL)	77.90 ± 7.55	72.06 ± 7.67	87.43 ± 7.47
HDL (mg/dL)	10.57 ± 1.70	19.90 ± 1.34 *	23.60 ± 6.09
Uric acid (mg/dL)	2.28 ± 0.34	1.40 ± 0.22	1.97 ± 0.36
Urea (mg/dL)	16.86 ± 1.40	14.00 ± 1.29	13.40 ± 1.14
Creatinine (mg/dL)	0.75 ± 0.09	1.00 ± 0.19	0.53 ± 0.15
Bilirubin (mg/dL)	0.18 ± 0.02	0.19 ± 0.02	0.10 ± 0.00 *
AST (UI)	150.57 ± 19.91	138.00 ± 27.06	160.67 ± 24.25
ALT (UI)	53.14 ± 9.66	69.67 ± 11.14	58.40 ± 11.71

Plasma biochemical parameters in standard diet (STD) group treated chronically over 15 days with N-acylethanolamines NAEs; oleoylethanolamide (OEA) and palmitoleylethanolamide (POEA). Data are expressed as means ± standard error of the mean (SEM) (*n* = 8 animals/group) analyzed by two-way ANOVA (diet and treatment) and Bonferroni post hoc test. (*) *p <* 0.05 significant differences compared with vehicle (VEH) group. VLDL; Very-low-density lipoprotein, LDL; low-density lipoprotein, HDL; hight-density lipoprotein, AST; aspartate aminotransferase, and ALT; alanine aminotransferase.

**Table 3 nutrients-13-02589-t003:** Plasma biochemical parameters from animals fed HFD and treated with either OEA or POEA (10 mg/kg) for 15 days.

Hypercaloric Cafeteria Diet (HFD)			
Plasma Metabolites	VEH	OEA	POEA
Glucose (mg/dL)	380.00 ± 9.08 ***	168.20 ± 17.11 ###	167.60 ± 8.28 ###
Triglycerides (mg/dL)	368.75 ± 82.68 **	173.00 ± 7.68 ###	198.00 ± 12.18 ##
Cholesterol (mg/dL)	211.00 ± 35.35 ***	75.20 ± 2.55 ###	79.40 ± 3.80 ###
VLDL (mg/dL)	73.75 ± 16.53 **	34.60 ± 1.54 ##	39.60 ± 4.44 ##
LDL (mg/dL)	106.82 ± 5.02 **	72.06 ± 7.67 ##	97.59 ± 5.20
HDL (mg/dL)	20.52 ± 0.95 ***	72.06 ± 7.67 ###	28,72 ± 2.94 #
Uric acid (mg/dL)	9.38 ± 1.95 ***	1.64 ± 0.25 ###	3.21 ± 0.90 ###
Urea (mg/dL)	72.92 ± 3.29 ***	33.75 ± 4.17 ###	43.30 ± 3.02 ###
Creatinine (mg/dL)	1.74 ± 0.12 **	0.69 ± 0.21 ###	0.87 ± 0.23 ##
Bilirubin (mg/dL)	0.99 ± 0.31 **	0.21 ± 0.03 ##	0.31 ± 0.12 ##
AST (UI)	553.00 ± 41.37 ***	288.33 ± 34.17 ##	400.40 ± 33.10 ##
ALT (UI)	206.33 ± 28.93 ***	85.00 ± 9.05 ###	140.40 ± 29.76

Plasma biochemical parameters in hypercaloric cafeteria diet (HFD) group treated chronically over 15 days with N-acylethanolamines NAEs; oleoylethanolamide (OEA) and palmitoleylethanolamide (POEA). Data are expressed as means ± standard error of the mean (SEM) (*n* = 8 animals/group) analyzed by two-way ANOVA (diet and treatment) and Bonferroni post hoc test. (**) *p* < 0.01 and (***) *p* < 0.001 significant differences compared with standard (STD) vehicle (VEH) group (Table 2); (#) *p <* 0.05, (##) *p <* 0.01, and (###) *p* < 0.001 significant differences compared with the VEH group. VLDL; Very-low-density lipoprotein, LDL; low-density lipoprotein, HDL; hight-density lipoprotein, AST; aspartate aminotransferase, and ALT; alanine aminotransferase.

## Data Availability

The data presented in this study are available on request from the corresponding authors.

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
