# Peer review of "Palmitoleoylethanolamide Is an Efficient Anti-Obesity Endogenous Compound: Comparison with Oleylethanolamide in Diet-Induced Obesity"

_nutrients, 2021, doi:10.3390/nu13082589_

Round 1

Reviewer 1 Report

The manuscript contains very interesting data indicating that N-aciletanolamides such as Oleoylethanolamide (OEA) and Palmitoleylethanolamide (POEA) can effect food intake and induce body mass reduction as well as improve many metabolic and inflammatory parameters.  However, this  manuscript needs significant improvement to be considered suitable for publication in a top scientific journal.

The text of this manuscript needs significant improved and must be corrected by a native English speaker familiar with biochemical sciences.

Eg. “plasmatic cholesterol” – should be plasma cholesterol or “biochemical parameters related to toxicity in the plasma´- what does it mean ? or  line 324: “Similar effect could be observed” – should be similar effects were observed  or  line 318 : “As we can observed in the VEH group, that kind a HFD produced significant negatives effects..” – there is hard to understand what “that kind”  mean?

It should be clarify if the observed changes in biochemical parameters are related first of all to changes in body mass of experimental animals or they result from specific action of OEA and POEA independent on the effect of this compounds on body mass.

Author Response

Dear Reviewer

We appreciate your suggestion. Following their recommendations, we have proceeded to make the suggested changes and corrections.

English language and style were extensively corrected by MDPI English Editing Service. In accordance with the second paragraph of your review, we have also reviewed and improved the main text with your recommendations. You can see them highlighted in blue font in the manuscript and indicated with comments in the right margin.

Finally, below we provide a detailed explanation to your comment in the third paragraph of your review now included in the Discussion (highlighting in blue);

“Our observations indicate that chronic administration of OEA and POEA induced body weight loss in rats under both STD and HFD feeding conditions. Interestingly, this slimming effect was more pronounced in the HFD-fed rats, especially in the group treated with OEA. Also, it was associated with a slight but significant anorectic effect of both NATs treatment in HFD-fed rats but not in STD-fed rats. Of note is that both the slimming and anorectic effects of OEA and POEA were also paralleled with a significant improvement of the plasma biochemical parameters with respect to the control group HFD-fed. Moreover, these biochemical parameters recovered levels very close to those found in STD-fed rats and vehicle treated. This effect could be due in part to the reduced food intake and the body mass observed in HFD-fed group. But even so, the tendency in reestablishing the plasma levels of the parameters evaluated is so robust that strongly indicates that NATs treatment exert a specific protective action on the integrity of liver and kidney tissues in obese rats. Indeed, the almost normal levels of bilirubin, ALT, HDL, creatinine, uric acid, etc., support this notion.”

Many tanks for you time

Dr. Juan Decara

Reviewer 2 Report

The manuscript describes the study that analyzes the influence of POEA and OEA treatment on the diet-induced animal model of obesity. The manuscript provides interesting findings based on substantial experimental work.

I have some minor tips regarding missing aspects in the manuscript.

  1. The introduction section is too long and should not contain discussion-like parts. Now it looks like a literature review article. Please short the introduction section to the more relevant data necessary to understand particular studied parameters and the aim of the study.
  2. Figure 1: Figure 1B as an experimental design should be moved to the Material and Methods section. In contrast, Figure 1 A may be replaced by explaining biochemical reactions directly in the manuscript's text.
  3. Some typo errors are in the manuscript's text and Figures (e.g., Fig. 9A, TD not STD, etc.).
  4. The meaning of asterisks in supplementary Fig. 1 is not given. 
  5. Fig. 4 chart is different in font size from others. Please improve the quality of the Figure.
  6. Western Blot results are presented in cut fragments. The better way to compare the intensity of particular bands is to separate all samples on the single gel with a reference protein. Is it possible to include an example of native results of the Western Blot in supplementary files?
  7. Did the authors use software to quantify oil red O stain quantification (e.g., staining area) in liver sections? If was used, the appropriate information should be put in the Materials and Methods section.
  8. The abstract should not exceed 200 words maximum; please check the length of the abstract carefully.

Author Response

Dear Reviewer

We appreciate your suggestion. Following their recommendations, we have proceeded to make the suggested changes and corrections.

English language and style were extensively corrected by MDPI English Editing Service.

Point 1: According your suggestion we shorten the Introduction section removing discussion-like parts and leaving only the most relevant references for our work. In addition, when removing some citations, the numbering of the references has also been modified (highlighted in green).

Point 2: We have replaced figure 1A by schematic of pathway and effects of POEA and POA, now being figure 1. Figure 1B was moved to the Material and Methods section according your suggestion, now being figure 2. Them, the numbering of the figures has also been modified (highlighted in blue).

Point 3: The main text was revised and typographical errors were corrected.

Point 4: The meaning of asterisks in supplementary Figure 1 were now explained in the figure foot.

Point 5: Figure 5 chart (previously figure 4) was improved, the fonts and size were matched to the other figures.

Point 6: We include all native results of the Western Blot in supplementary files (Figure S2 and S3). Note that both the images of membranes with Punseau S solution and the gels include a LEA group that was not included in the present study. This group is included in the figures for not to modify the original images.

Point 7: in this new version of the manuscript, we include the complete information to quantify oil red O stain (highlighted in blue).

Point 8: we have checked the number of words in the abstract and we have adjusted it to 200 words of length.

Many tanks for you time

Dr. Juan Decara

Round 2

Reviewer 1 Report

The manuscript was found significantly imprpved according to the previous commments.

No additional comments